# Robotic versus open component separation: A retrospective cohort and propensity score analysis of complication rates and clinical outcomes

Silvana Maldonado[1]*, Carlos Gonzales[1]*, Anshumi Desai[2], Jiddu Guart[3], Alba Zevallos Ventura[4], Bryan Valcarcel[5], Joseph Escandon[6], Natalia Mejia[7], Camila Franco Mesa[8], J. Smith Torres Roman[9], Hamed Sarikhani[10], Jose Luis Barrueto[1], Gabriel De la Cruz Ku[1], Donald Czerniach[3], Sameer Patel[7], Adam Walchak[7]

**1** Universidad Cientifica del Sur, Lima, Peru, **2** Department of Plastic Surgery, University of Miami Miller School of Medicine, Miami, Florida, United States of America, **3** University of Massachusetts Medical School, Worcester, Massachusetts, United States of America, **4** Department of Surgery, Loma Linda University, Loma Linda, California, United States of America, **5** Department of Lymphoma Myeloma, The University of Texas MD Anderson Cancer Center, Houston, Texas, United States of America, **6** Department of Surgery, Wyckoff Heights Medical Center, Brooklyn, New York, United States of America, **7** Department of Plastic and Reconstructive Surgery, Temple University, Philadelphia, Pennsylvania, United States of America, **8** Department of Surgery, University of Texas Medical Branch at Galveston, Galveston, Texas, United States of America, **9** Escuela de Posgrado, Universidad Tecnologica del Peru, Lima, Peru, **10** Department of Emergency Medicine, Montefiore Medical Center, New York, New York, United States of America

* gabrieldelacruzku@gmail.com

## Abstract

### Background

Complex ventral hernias are a surgical challenge associated with high morbidity and healthcare costs. Component separation techniques have improved throughout the years with better outcomes, although the optimal approach remains debated. Robotic surgery has shown promising outcomes as an alternative to open repair, although data in large multicenter studies is still limited.

### Methods

A retrospective cohort study was conducted using the American College of Surgeons National Surgical Quality Improvement Program (ACS-NSQIP) database. Adult patients undergoing component separation for ventral hernia repair were identified using CPT and ICD codes. Outcomes included 30-day surgical, wound, medical, and overall complications, as well as length of stay and readmission. Multivariable logistic regression and propensity score matching were applied to adjust for baseline differences.

**Data availability statement:** All relevant data are within the manuscript and its Supporting information file (S1 File).

**Funding:** The author(s) received no specific funding for this work.

**Competing interests:** The authors have declared that no competing interests exist.

## Results

A total of 6,207 patients were included, from those 4,443 (71.6%) underwent open technique and 1,764 (28.4%) robotic. After propensity matching (n = 5,259), robotic repair was independently associated with significantly lower overall complication rates (4.8% vs. 19.6%, aOR 0.193, 95% CI 0.140–0.265, p < 0.001), including wound (2.2% vs. 10.2%, aOR 0.164, p < 0.001), surgical (2.9% vs. 10.0%, aOR 0.271, p < 0.001), and medical complications (2.0% vs. 7.0%, aOR 0.229, p < 0.001). Robotic surgery was also associated with shorter length of stay (1.34 vs. 3.86 days, p < 0.001) and lower readmission rates (4.4% vs. 9.1%, p < 0.001).

## Conclusions

Robotic component separation for ventral hernia repair is associated with lower postoperative complication rates, shorter length of stay, and fewer readmissions compared to the open approach. These benefits remained significant after multivariate analysis and propensity score matching, supporting the robotic technique as an effective strategy. Prospective studies are warranted to evaluate long-term outcomes, including recurrence, and to assess cost-effectiveness to optimize evidence-based surgical decision-making.

## Introduction

Ventral abdominal hernias are protrusions of intra-abdominal contents through a defect in the anterior abdominal wall, excluding inguinal and diaphragmatic hernias [1,2]. Their classification is clinically relevant, as it influences risk assessment, therapeutic planning, and outcomes [1]. They may be of primary or secondary origin; the latter is often represented by incisional hernias [2]. There are many proposals for classification systems; however, this paper will focus on complex ventral hernias, also referred to as "giant" ventral hernias. These are typically defined as large fascial defects greater than 10 cm or as having a bilateral rectus width-to-hernia defect ratio of less than 2:1 [3,4].

The global prevalence of ventral hernias is estimated at approximately 5%, with approximately 25,000 and 600,000 repairs in the United Kingdom and the United States, respectively [2,5]. These figures underscore that ventral hernias represent not only a frequent surgical condition but also a significant public health burden. In the United States, the annual healthcare expenditure related to ventral hernia management is estimated at approximately $9.7 billion, nearly triple the $3.2 billion reported in earlier studies [6,7] Costs and outcomes are strongly influenced by patient comorbidities and postoperative complications, with reported inpatient per-patient costs reaching up to $80,000 and average expenditures ranging from $20,000 to $35,000 [6,8–11].

The management of ventral hernias is crucial, as untreated hernias are associated with complications and negatively impact psychological well-being and quality

of life [12,13]. In complex ventral hernias, surgical management has evolved toward component separation techniques, which involve mobilizing the muscular components of the abdominal wall to achieve tension-free closure, thereby avoiding wound-related complications caused by excessive tension, generally with prosthetic mesh reinforcement [14,15]. Among these, anterior component separation, first described by Ramirez et al. in 1990, involves incising the external oblique aponeurosis lateral to the linea semilunaris and dissecting the plane between the external and internal oblique muscles, thereby allowing medial advancement of the rectus abdominis muscles for tension-free midline closure [4]. On the other hand, posterior component separation includes the Rives-Stoppa technique and the Transversus Abdominis Release (TAR), introduced by Novitsky et al. in 2012, which has gained wide acceptance and can be performed via laparotomy or minimally invasive approaches (laparoscopy or robotic surgery) [16–18]. Currently, TAR is considered the preferred technique for many complex hernias due to its ability to achieve wide medialization and its association with low recurrence rates, even in high-risk patients with comorbidities such as obesity, advanced age, or elevated BMI [19–21].

Today, component separation is considered the technique of choice for the repair of complex ventral hernias. It can be performed through an open or minimally invasive approach, using robotic or laparoscopic techniques. The laparoscopic approach, however, is less frequently adopted due to suboptimal ergonomics, which may translate into longer operative times despite low perioperative complication rates and shorter hospital stay [21]. Robotic surgery has emerged as an alternative that preserves the advantages of minimally invasive surgery while overcoming some of these limitations.

A recent systematic review and meta-analysis by Bracale et al. demonstrated that robotic is associated with fewer systemic complications and shorter length of stay compared to the open approach, with no significant differences in readmission or reoperation rates. However, the included studies were limited to small retrospective series with significant heterogeneity, highlighting the need for larger multicenter analyses with adequate adjustment [22]. To date, there is limited data from large-scale, multicenter analyses incorporating patient-level risk adjustment. The present study aims to address this gap by utilizing the American College of Surgeons – National Surgical Quality Improvement Program (ACS – NSQIP) database, which provides prospectively collected, standardized, and validated data from multiple hospitals across the United States. Using multivariable regression and propensity score matching to minimize selection bias, this study aims to provide more generalizable evidence to guide clinical decision-making in the surgical management of complex ventral hernias.

## Materials & methods

### Database and study design

A retrospective cohort study was conducted, using the American College of Surgeons National Surgical Quality Improvement Program (ACS-NSQIP). This is a de-identified, prospectively collected clinical dataset from adult surgical patients at participating institutions, obtained through a systematic randomized sampling protocol. The study was approved by the Institutional Review Board at Universidad Cientifica del Sur, Lima, Peru. To ensure a comprehensive identification of complex cases, a double-coding strategy was employed. ICD-10 codes K43.2 (incisional hernia without obstruction or gangrene) and K43.9 (unspecified ventral hernia) were used to capture a broad population of patients with ventral wall defects, including both incisional and spontaneous ventral hernias. Cases involving component separation were then identified using CPT codes 15734 and 49659; for the latter, the NSQIP robotic-assisted variable was used to confirm the robotic approach. This strategy allowed the study population to be defined not only by the hernia diagnosis but also by the corresponding surgical procedure. It should be noted that the available coding does not differentiate between anterior and posterior component separation techniques. Patients were excluded if they met any of the following criteria: age < 18 years old, emergency procedures, laparoscopic approach, procedures not meeting CPT or International Classification of Diseases (ICD) code criteria, non-ventral hernia diagnosis, re-operative repairs.

### Definition of variables and classification of cohorts

Patients were categorized into two cohorts based on the surgical approach used to perform component separation: open versus robotic. Procedures were identified using the CPT and ICD codes described above.

### Outcomes

All the complications were assessed within 30 days after the index surgery. Data was accessed from October 25th, 2025 to November 15th, 2025. Complications with frequency zero were excluded from the analysis. Wound complications were defined as superficial surgical site infection (SSI), deep surgical site infection, organ/space infections, and wound dehiscence. Operative complications were defined as returning to the operating room and bleeding that required intra- or postoperative transfusion. Medical complications included pneumonia, stroke, myocardial infarction, deep vein thrombosis, and postoperative sepsis. Overall complications included all the mentioned above. In addition, clinical, pathological, and surgical characteristics were compared between cohorts in relation to the surgical approach.

### Statistical analysis

Descriptive statistics were used to summarize demographic and clinical variables. Absolute and relative frequencies were used for categorical variables, while mean with standard deviation and median with standard deviation were used for quantitative variables depending on the distribution of the variable. The normality of the quantitative variables was assessed with the Shapiro-Wilk test. For univariate analyses, chi-square was used to analyze the relation between two categorical variables, while student-t test and Mann-Whitney U test were used for categorical variables with quantitative variables with normal and abnormal distribution, respectively. To minimize selection bias and adjust for baseline differences between groups, Propensity Score Matching (PSM) was performed. Propensity scores were estimated using a logistic regression model, with surgical technique as the dependent variable and preoperative demographic, clinical, and surgical variables as covariates. The variables used to balance the propensity score model included age, albumin, BMI, diabetes mellitus, sex, and smoking status.

Matching was performed in a 1:2 ratio using the nearest neighbor method, with a caliper of 0.2 standard deviations of the logit of the propensity score. Covariate balance between matched groups was assessed using the standardized mean difference (SMD), with values <0.1 indicating acceptable balance (Fig 1). Patients were matched on age, sex, smoking status, body mass index, diabetes mellitus, and serum albumin level.

Finally, binary logistic regression was performed on both matched and unmatched cohorts to evaluate the association between surgical technique and postoperative complications. In the matched cohort, this followed a doubly robust estimation approach by including the matching variables as covariates in the model to account for any residual imbalance. Results were reported as odds ratios (OR) with 95% confidence intervals (CI). A p-value < 0.05 was considered statistically significant.

### Ethical considerations

This study was approved by the Institutional Review Board of the Universidad Científica del Sur (approval number: PRE-15-2025-00460). Due to the retrospective design of the study and the use of fully de-identified data, the requirement for informed consent was waived by the ethics committee. No direct patient contact occurred, and no identifiable information was accessed by the investigators. All data were handled confidentially and used exclusively for research purposes. Access to the encoded dataset was restricted to the research team.

### Results

Of the 6,791 patients who underwent component separation from 2022 to 2023, 6,207 met the inclusion criteria. Among these, 4,443 (71.6%) underwent open repair, while 1,764 (28.4%) underwent robotic repair. Patients undergoing open

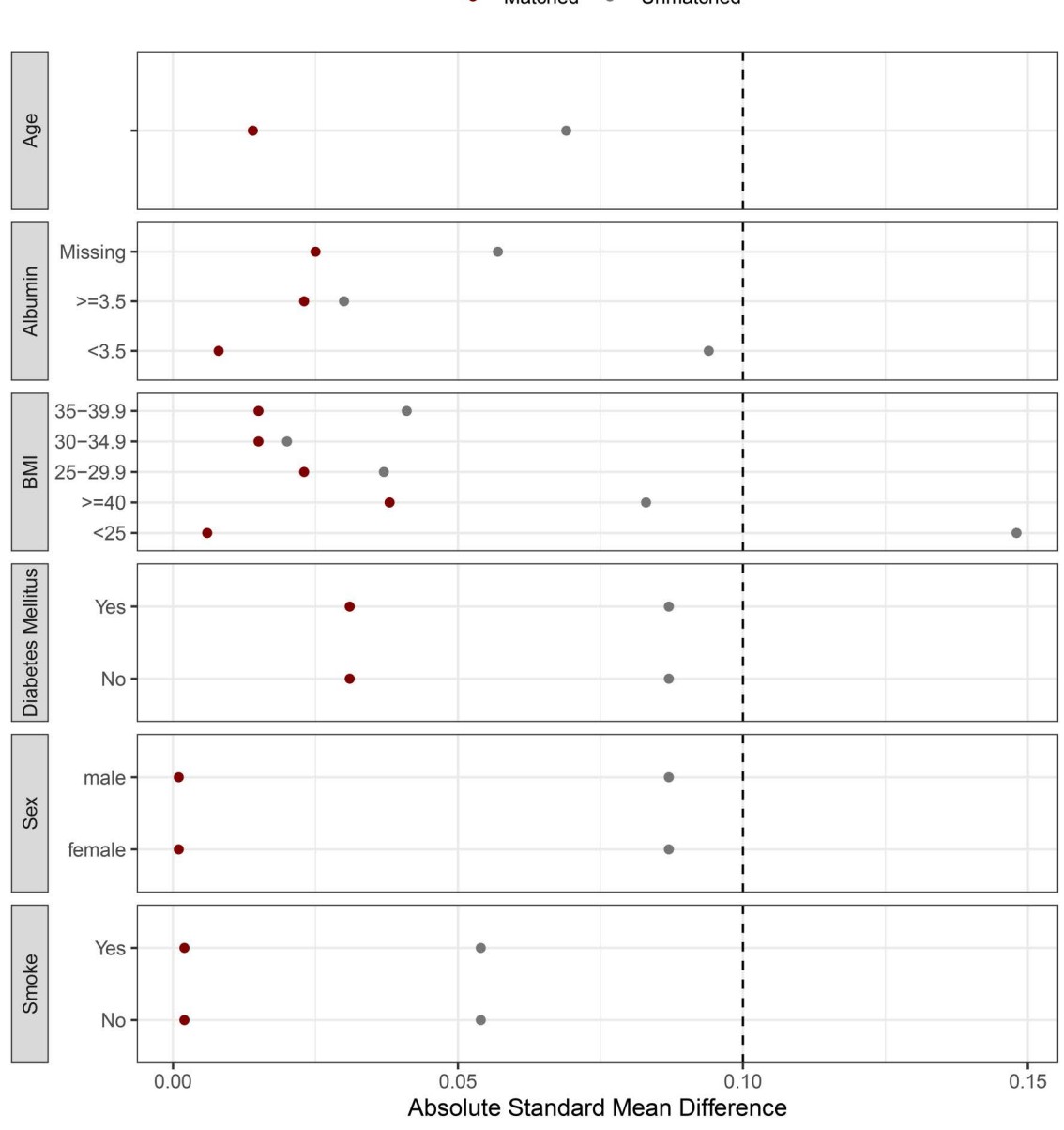

**Fig 1. Covariate balance after propensity score matching for key preoperative variables.**

surgery were slightly older compared to those in the robotic group (mean age 59.48 ± 12.57 vs. 58.59 ± 12.96 years, p = 0.013). The proportion of patients aged >65 years was also more prevalent in the open group (1,725 [38.8%] vs. 636 [36.0%], p = 0.041). Females were significantly more common in the open surgery group (2,442 [55.0%]) compared to robotic surgery (890 [50.5%], p = 0.001). Hispanic ethnicity was notably higher in robotic surgeries (231 [13.1%]) versus open surgeries (482 [10.8%], p < 0.001). Significant differences were observed in specialty, BMI, diabetes mellitus, congestive heart failure (CHF), hypertension, immunosuppression, ASA classification, hematocrit, and albumin levels (p < 0.05). After Propensity matching, the sample consisted of 5,259 patients (open: 3,506; robotic: 1,753). Age, sex, race, and BMI distributions became comparable between groups (all p > 0.05). All matched variables achieved

a Standardized Mean Difference (SMD) <0.1, indicating adequate balance Hispanic ethnicity remained significantly higher in the robotic group (231 [13.2%]) compared to open surgery (391 [11.2%], p<0.001). Robotic surgeries had exclusively general surgery specialists (1,753 [100%]) versus open surgery (1,411 [89.7%], p<0.001). Congestive heart failure remained significantly more common in open surgery (128 [3.7%]) compared to robotic (39 [2.2%], p=0.005). Immunosuppression was significantly lower in robotic surgery patients (111 [6.3%]) compared to open surgery (347 [9.9%], p<0.001). Significant differences persisted in ASA classification (p<0.001) and hematocrit values (p=0.018). (Table 1).

Among 6,207 patients, before propensity matching, at 30-day follow-up, the overall complication rate was significantly lower in the robotic group (84 [4.8%]) compared to the open group (877 [19.7%], p<0.001). Surgical complications occurred in 452 (10.2%) open cases and 50 (2.8%) robotic cases (p<0.001). Wound complications (445 [10.0%] vs. 39 [2.2%], p<0.001), medical complications (309 [7.0%] vs. 35 [2.0%], p<0.001), and superficial surgical site infections (SSI) (234 [5.3%] vs. 19 [1.1%], p<0.001) were all significantly more frequent in the open group. Other notable differences favoring the robotic approach included reduced rates of deep SSI (96 [2.2%] vs. 8 [0.5%], p<0.001), organ/space SSI (97 [2.2%] vs. 13 [0.7%], p<0.001), and wound dehiscence (34 [0.8%] vs. 1 [0.1%], p=0.001). Pulmonary and thromboembolic events such as pneumonia (102 [2.3%] vs. 12 [0.7%], p<0.001), reintubation (54 [1.2%] vs. 5 [0.3%], p=0.003), pulmonary embolism (64 [1.4%] vs. 6 [0.3%], p<0.001), and DVT (52 [1.2%] vs. 6 [0.3%], p=0.002) were also significantly less frequent in the robotic cohort. In the propensity matched cohort of 5,259 patients (open: 3,506; robotic: 1,753), robotic surgery continued to demonstrate significantly lower overall complication rates (84 [4.8%] vs. 688 [19.6%], p<0.001). Surgical (50 [2.9%] vs. 350 [10.0%], p<0.001) and wound complications (39 [2.2%] vs. 356 [10.2%], p<0.001) remained markedly reduced in the robotic group. Rates of superficial SSI (19 [1.1%] vs. 183 [5.2%], p<0.001), deep SSI (8 [0.5%] vs. 78 [2.2%], p<0.001), and organ/space SSI (13 [0.7%] vs. 82 [2.3%], p<0.001) continued to favor the robotic approach. Medical complications (35 [2.0%] vs. 247 [7.0%], p<0.001), pneumonia (12 [0.7%] vs. 76 [2.2%], p<0.001), and ventilator dependence >48 hours (2 [0.1%] vs. 38 [1.1%], p<0.001) were all less frequent with robotic surgery. Other outcomes such as bleeding requiring transfusion (11 [0.6%] vs. 111 [3.2%], p<0.001), sepsis (13 [0.7%] vs. 52 [1.5%], p=0.022), and return to the operating room (25 [1.4%] vs. 162 [4.6%], p<0.001) were also significantly lower in the robotic group. Mean length of stay was shorter following robotic surgery (1.34±3.16 vs. 3.86±8.42 days, p<0.001). Readmission rates were significantly lower for robotic cases (78 [4.4%]) compared to open (318 [9.1%], p<0.001). (Table 2).

In the unadjusted analysis of the unmatched cohort, robotic surgery was associated with significantly lower odds of overall complications (OR 0.203, 95% CI 0.161–0.256, p<0.001), wound complications (OR 0.203, 95% CI 0.146–0.283, p<0.001), surgical complications (OR 0.258, 95% CI 0.191–0.347, p<0.001), and medical complications (OR 0.271, 95% CI 0.190–0.386, p<0.001). These associations remained statistically significant in multivariate analysis, including for overall complications (aOR 0.206, 95% CI 0.150–0.282, p<0.001) and across all complication categories. Other independent predictors of higher complication rates in multivariate analysis included age, BMI, ASA classification, hypoalbuminemia (<3.5), CHF, and COPD. Smoking was associated with increased risk for overall and wound complications. Diabetes was not independently associated with higher complication risk after adjustment. (Table 3).

After propensity score matching, which achieved balance across age, albumin, BMI, diabetes mellitus, sex, and smoking status based on standardized mean differences (SMD<0.1). Robotic surgery remained strongly associated with reduced complications (Fig 1). In multivariate analysis, robotic surgery was independently protective for overall (aOR 0.193, 95% CI 0.140–0.265, p<0.001), wound (aOR 0.164, 95% CI 0.100–0.269, p<0.001), surgical (aOR 0.271, 95% CI 0.182–0.402, p<0.001), and medical complications (OR 0.229, 95% CI 0.135–0.389, p<0.001). Other significant predictors of increased complication risk included higher BMI, ASA classification, low serum albumin, CHF, and COPD. Smoking remained significantly associated with overall and wound complications. Similar to the unmatched analysis, diabetes mellitus was not an independent predictor of complications in the adjusted models. (Table 4).

 

**Table 1. Clinicopathological and surgical characteristics of patients who underwent component abdominal separation from 2022 to 2023 according to surgical approach.**

| Characteristics | Before propensity matching | | | | After propensity matching | | | |
|---|---|---|---|---|---|---|---|---|
| | Surgical approach | | Total N=6207 | P value | Surgical approach | | Total N=5259 | P value |
| | Open N=4443 | Robotic N=1764 | | | Open N=3506 | Robotic N=1753 | | |
| Age – years ± SD | 59.48 ± 12.57 | 58.59 ± 12.96 | 59.22 ± 12.69 | 0.013 a | 58.77 ± 12.69 | 58.59 ± 12.96 | 58.71 ± 12.78 | 0.629 a |
| Age > 65 years | 1725 (38.8) | 636 (36.0) | 2361 (38.0) | 0.041 b | 1297 (37.0) | 631 (64.0) | 1928 (36.7) | 0.479 b |
| Sex | | | | | | | | |
| Female | 2442 (55.0) | 890 (50.5) | 3332 (53.7) | 0.001 b | 1772 (50.5) | 887 (50.6) | 2659 (50.6) | 0.969 b |
| Male | 2001 (45.0) | 874 (49.5) | 2875 (46.3) | | 1734 (49.5) | 866 (49.4) | 2600 (49.4) | |
| Race | | | | | | | | |
| White | 3460 (84.3) | 1328 (82.9) | 4788 (83.9) | 0.526 b | 2721 (84.0) | 1320 (82.8) | 4041 (83.6) | 0.730 b |
| African-American | 459 (11.2) | 201 (12.5) | 660 (11.6) | | 373 (11.5) | 200 (12.5) | 573 (11.9) | |
| Asian | 52 (1.3) | 19 (1.2) | 71 (1.2) | | 40 (1.2) | 19 (1.2) | 59 (1.2) | |
| Other | 135 (3.3) | 55 (3.4) | 190 (3.3) | | 105 (3.2) | 55 (3.5) | 160 (3.3) | |
| Hispanic ethnicity | 482 (10.8) | 231 (13.1) | 713 (11.5) | <0.001 b | 391 (11.2) | 231 (13.2) | 622 (11.8) | <0.001 b |
| Specialty | | | | | | | | |
| General Surgery | 3969 (99.4) | 1765 (100.0) | 5734 (92.4) | <0.001 b | 1411 (89.7) | 1753 (100.0) | 3164 (60.2) | <0.001 b |
| Plastic Surgery | 472 (10.6) | 0 (0.0) | 473 (7.6) | | 361 (10.3) | 0 (0.0) | 361 (39.8) | |
| BMI – kg/m2 | 32.18 ± 6.26 | 33.17 ± 6.27 | 32.46 ± 6.28 | <0.001 a | 32.77 ± 6.19 | 33.16 ± 6.26 | 32.91 ± 6.22 | 0.289 a |
| BMI classification | | | | | | | | |
| Normal | 489 (11.1) | 127 (7.2) | 616 (10.0) | <0.001 b | 259 (7.4) | 127 (7.2) | 389 (7.3) | 0.656 b |
| Overweight | 1227 (27.8) | 458 (26.1) | 1685 (27.3) | | 954 (27.2) | 459 (26.2) | 1413 (26.9) | |
| Obesity type 1 | 1396 (31.6) | 572 (32.6) | 1968 (31.9) | | 1167 (33.3) | 571 (32.6) | 1738 (33.0) | |
| Obesity type 2 | 846 (19.2) | 365 (20.8) | 1211 (19.6) | | 709 (20.2) | 365 (20.8) | 1074 (20.4) | |
| Obesity type 3 | 457 (10.4) | 231 (13.2) | 688 (11.2) | | 417 (11.9) | 231 (13.2) | 648 (12.3) | |
| Smoking | 567 (12.8) | 194 (11.0) | 761 (12.3) | 0.055 b | 386 (11.0) | 194 (11.1) | 580 (11.0) | 0.950 b |
| Diabetes Mellitus | 992 (22.3) | 334 (18.9) | 1326 (21.4) | 0.003 b | 621 (17.7) | 332 (18.9) | 953 (18.1) | 0.276 b |
| COPD | 218 (4.9) | 75 (4.2) | 292 (4.7) | 0.232 b | 155 (4.4) | 74 (4.2) | 229 (4.4) | 0.738 b |
| CHF | 181 (4.1) | 40 (2.3) | 221 (3.6) | 0.001 b | 128 (3.7) | 39 (2.2) | 167 (3.2) | 0.005 b |
| Hypertension | 2470 (55.6) | 917 (52.0) | 3387 (54.6) | 0.010 b | 1894 (54.0) | 915 (52.2) | 2809 (53.4) | 0.211 b |
| Metastatic cancer | 54 (1.2) | 14 (0.8) | 68 (1.1) | 0.149 b | 40 (1.1) | 14 (0.8) | 54 (1.0) | 0.246 b |
| Immunosuppression | 445 (10.0) | 113 (6.4) | 558 (9.0) | <0.001 b | 347 (9.9) | 111 (6.3) | 458 (8.7) | <0.001 b |
| Bleeding disorder | 146 (3.3) | 42 (2.4) | 188 (3.0) | | 107 (3.1) | 43 (2.5) | 150 (2.9) | 0.219 b |
| ASA – ASA classification ± SD | 2.64 ± 0.55 | 2.53 ± 0.57 | 2.61 ± 0.55 | <0.001 a | 2.64 ± 0.55 | 2.53 ± 0.57 | 2.60 ± 0.56 | <0.001 a |
| Hematocrit – % ± SD | 40.94 ± 4.62 | 41.54 ± 4.36 | 41.11 ± 4.56 | <0.001 a | 41.18 ± 4.63 | 41.54 ± 4.36 | 41.30 ± 4.55 | 0.018 a |
| Platelets – platelets/mcL ± SD | 251.69 ± 79.12 | 249.53 ± 70.95 | 251 ± 0.76.98 | 0.378 a | 225.12 ± 0.78 | 249.82 ± 71.00 | 250.02 ± 76.05 | 0.906 a |
| Albumin <3.5 | 154 (6.1) | 37 (3.9) | 191 (5.5) | 0.010 b | 79 (4.1) | 36 (3.8) | 115 (4.0) | 0.727 b |
| Creatinine – mg/dL ± SD | 0.99 ± 0.38 | 0.99 ± 0.45 | 0.99 ± 0.40 | 0.617 a | 1.00 ± 0.40 | 0.99 ± 0.46 | 0.99 ± 0.42 | 0.420 a |
| Operative time – min ± SD | 210.84 ± 101.49 | 213.41 ± 102.41 | 211.56 ± 101.875 | 0.377 a | 207.58 ± 101.71 | 206.50 ± 105.14 | 207.22 ± 102.86 | 0.718 a |

COPD, chronic obstructive pulmonary disease; CHF, chronic heart failure; ASA, American Society of Anesthesiologists. Analysis 1, comparison between Modified 5-item Frailty Index: 0, 1, ≥2; Analysis 2, comparison between Modified 5-item Frailty Index <2, ≥2.

a p-value calculated using Student's T test.

b p-value calculated using chi-square test.

Table 2. Post-operative complications of patients who underwent oncoplastic surgery from 2005 to 2020 according to surgical approach.

| Characteristics | Before propensity matching | | | | After propensity matching | | | |
|---|---|---|---|---|---|---|---|---|
| | Surgical approach | | Total N = 6207 | P value | Surgical approach | | Total N = 5259 | P value |
| | Open N = 4443 | Robotic N = 1764 | | | Open N = 3506 | Robotic N = 1753 | | |
| Overall complications | 877 (19.7) | 84 (4.8) | 961 (15.5) | <0.001 [a] | 688 (19.6) | 84 (4.8) | 772 (14.7) | <0.001 [a] |
| Surgical complications | 452 (10.2) | 50 (2.8) | 502 (8.1) | <0.001 [a] | 350 (10.0) | 50 (2.9) | 400 (7.6) | <0.001 [a] |
| Wound complications | 445 (10.0) | 39 (2.2) | 848 (7.8) | <0.001 [a] | 356 (10.2) | 39 (2.2) | 395 (7.5) | <0.001 [a] |
| Medical complications | 309 (7.0) | 35 (2.0) | 344 (5.5) | <0.001 [a] | 247 (7.0) | 35 (2.0) | 282 (5.4) | <0.001 [a] |
| Superficial SSI | 234 (5.3) | 19 (1.1) | 253 (4.1) | <0.001 [a] | 183 (5.2) | 19 (1.1) | 202 (3.8) | <0.001 [a] |
| Deep SSI | 96 (2.2) | 8 (0.5) | 104 (1.7) | <0.001 [a] | 78 (2.2) | 8 (0.5) | 86 (1.6) | <0.001 [a] |
| Organ/space SSI | 97 (2.2) | 13 (0.7) | 110 (1.8) | <0.001 [a] | 82 (2.3) | 13 (0.7) | 95 (1.8) | <0.001 [a] |
| Wound dehiscence | 34 (0.8) | 1 (0.1) | 35 (0.6) | 0.001 [a] | 26 (0.7) | 1 (0.1) | 27 (0.5) | 0.001 [a] |
| Pneumonia | 102 (2.3) | 12 (0.7) | 114 (1.8) | <0.001 [a] | 76 (2.2) | 12 (0.7) | 88 (1.7) | <0.001 [a] |
| Reintubation | 54 (1.2) | 5 (0.3) | 59 (0.9) | 0.003 [a] | 38 (1.1) | 5 (0.3) | 43 (0.8) | 0.008 [a] |
| Pulmonary embolism | 64 (1.4) | 6 (0.3) | 70 (1.1) | <0.001 [a] | 50 (1.4) | 6 (0.3) | 56 (1.1) | <0.001 [a] |
| Deep vein thrombosis | 52 (1.2) | 6 (0.3) | 58 (0.9) | 0.002 [a] | 39 (1.1) | 6 (0.3) | 45 (0.9) | 0.004 [a] |
| >48 on the ventilator | 49 (1.1) | 2 (0.1) | 51 (0.8) | <0.001 [a] | 38 (1.1) | 2 (0.1) | 40 (0.8) | <0.001 [a] |
| Renal insufficiency | 176 (4.0) | 9 (0.5) | 185 (3.0) | <0.001 [a] | 138 (3.9) | 9 (0.5) | 147 (2.8) | <0.001 [a] |
| Urinary tract infection | 57 (1.3) | 10 (0.6) | 67 (1.1) | 0.014 [a] | 39 (1.1) | 10 (0.6) | 49 (0.9) | 0.054 [a] |
| Stroke | 7 (0.2) | 0 (0.0) | 7 (0.1) | 0.095 [a] | 5 (0.1) | 0 (0.0) | 5 (0.1) | 0.114 [a] |
| Cardiac arrest | 23 (0.6) | 2 (0.1) | 25 (0.4) | 0.072 [a] | 19 (0.6) | 2 (0.1) | 21 (0.4) | 0.063 [a] |
| Myocardial infarction | 25 (0.6) | 4 (0.2) | 29 (0.5) | 0.080 [a] | 17 (0.5) | 4 (0.2) | 21 (0.4) | 0.164 [a] |
| Bleeding requiring blood transfusion | 144 (3.2) | 11 (0.6) | 155 (2.5) | <0.001 [a] | 111 (3.2) | 11 (0.6) | 122 (2.3) | <0.001 [a] |
| Post operative sepsis | 66 (1.5) | 13 (0.7) | 79 (1.3) | 0.018 [a] | 52 (1.5) | 13 (0.7) | 65 (1.2) | 0.022 [a] |
| Septic shock | 40 (0.9) | 5 (0.3) | 45 (0.7) | 0.010 [a] | 30 (0.9) | 5 (0.3) | 35 (0.7) | 0.016 [a] |
| Length of stay | 3.76±3.24 | 1.36±3.24 | 3.08±8.20 | <0.001 [b] | 3.86±8.424 | 1.34±3.161 | 3.02±7.21 | <0.001 [b] |
| Return to the operating room | 198 (4.5) | 25 (1.4) | 223 (3.6) | <0.001 [a] | 162 (4.6) | 25 (1.4) | 187 (3.6) | <0.001 [a] |
| Readmission | 405 (9.1) | 78 (4.4) | 483 (7.8) | <0.001 [a] | 318 (9.1) | 78 (4.4) | 396 (7.5) | <0.001 [a] |

SSI, surgical site infection.

a p-value calculated using chi-square test.

b p-value calculated using Student's T test.

## Discussion

Our analysis of the NSQIP data demonstrates that robotic component separation is associated with improved short-term outcomes compared to the open approach. Patients undergoing robotic repair experienced significantly lower overall postoperative morbidity and shorter hospital stays. Our findings showed an overall complication rate of 4.8% in the robotic group compared to 19.6% in the open surgery group. These findings reinforce the growing trend in complex hernia surgery favoring minimally invasive techniques, as recent studies have similarly found that robotic abdominal wall reconstruction can reduce complication rates and hasten recovery [19,22].

In this study, open component separation had a markedly higher complication rate than the robotic approach, both before and after propensity score matching. Before matching, surgical complications occurred in 10.2% of open cases versus 2.8% of robotic cases, and after matching, these were 10.0% versus 2.9%, respectively. Similarly, wound

complications remained significantly lower with robotic surgery (2.2% vs 10.2%). Propensity matching balanced baseline characteristics, yet the robotic cohort still showed significantly fewer 30-day complications, explaining a true reduction in risk with robotic surgery. This mirrors prior comparisons of robotic versus open component separation. Dewulf et al. reported that open component separation had over twice the rate of serious complications (Clavien–Dindo grade ≥III) and surgical-site infections compared to the robotic technique [23]. Similarly, Martin-Del-Campo and colleagues noted virtually no major systemic complications in their robotic component separation group versus 17% in the open approach, alongside a substantial drop in wound morbidity and length of stay with the robotic technique [24] Bittner et al. also observed roughly a 50% reduction in overall morbidity with the robotic approach (appro. ximately 19% vs 39%) and a halving of hospital length of stay (median ~3 days vs 6 days) relative to open surgery [25]. These consistent reductions in complication rates highlight the clinical benefit of performing component separation robotically.

The multivariable analysis further confirmed an independent association between surgical approach and outcomes: open component separation carried significantly higher odds of postoperative complications than the robotic approach, even after adjusting for confounders. Our study indicated robotic surgery substantially reduced the odds of overall (OR 0.206), wound (OR 0.203), surgical (OR 0.258), and medical complications (OR 0.271). The robotic technique appears to offer advantages directly affecting surgical outcomes. Several factors likely account for these better outcomes with robotic surgery. First, the minimally invasive nature of robotic component separation avoids the large skin incisions and extensive subcutaneous dissection required in open repair, thereby preserving perfusion to the abdominal wall and limiting dead space [26]. Ghali et al. demonstrated that a limited-access component separation significantly lowered wound complication rates, which they attributed to preserved skin vascularity and reduced tissue trauma [26]. Moreover, the robotic platform provides superior visualization and instrument dexterity. The three-dimensional optics and wristed instruments enable precise dissection and fascial closure with minimal collateral damage [27]. This improved surgical precision translates into less intraoperative blood loss and tissue injury – as evidenced by one study showing the robotic component separation group had only one-third the blood loss of the open group [24] and likely a lower systemic inflammatory response. Collectively, these advantages (reduced tissue trauma and better visualization) logically lead to fewer wound-healing problems and other complications in the postoperative period.

After propensity matching, robotic surgery maintained superior outcomes in our cohort, demonstrating significantly lower rates of overall complications (OR 0.193), wound complications (OR 0.164), surgical complications (OR 0.271), and medical complications (OR 0.229). Additionally, robotic patients had markedly shorter hospital stays (mean of 1.34 vs 3.86 days) and reduced readmission rates (4.4% vs 9.1%). Notably, our matched analysis aligns with other risk-adjusted comparisons in the literature. Henriksen et al. recently reported in a nationwide matched study that robot-assisted ventral hernia repair was associated with a dramatically shorter length of stay (on average, half a day vs 2 days) and reduced 90-day readmissions compared to open repair [28]. A systematic review and meta-analysis by Bracale et al. [22] similarly concluded that robotic component separation improves recovery and appears to decrease postoperative complications relative to open surgery. The consistently better outcomes observed with robotic component separation can be explained by the factors discussed above. By minimizing soft-tissue disruption and preserving abdominal wall blood supply, the robotic technique mitigates the risk of wound infection, dehiscence, and seroma formation. Additionally, patients benefit from less postoperative pain and earlier mobilization, contributing to faster recoveries and fewer downstream complications [27].

Strengths of this study include the use of a large, multicenter NSQIP database, which enhances the generalizability of the findings. The total cohort included 6,791 patients providing a robust sample for analysis. To minimize selection bias and account for confounders, propensity score matching was performed, resulting in 1753 patients in the robotic group and 3,506 in the open group, with well-balanced baseline characteristics. These methodological approaches strengthen the validity of the comparisons between robotic and open component separation.

**Table 3. Univariate and multivariate analysis for overall, medical, wound, and surgical complications before propensity score matching, 2022-2023.**

| Characteristics | Overall complications | | | | | | Wound complications | | | | | |
|---|---|---|---|---|---|---|---|---|---|---|---|---|
| | Univariate | | | Multivariate | | | Univariate | | | Multivariate | | |
| | OR | 95%CI | P value | OR | 95%CI | P value | OR | 95%CI | P value | OR | 95%CI | P value |
| Age - years | 1.009 | 1.004-1.015 | 0.001 | 1.009 | 1.000-1.018 | 0.046 | 0.993 | 0.986-1.000 | 0.048 | 0.988 | 0.978-1.000 | 0.042 |
| Sex | | | | | | | | | | | | |
| Female | 1.00 | | | 1.00 | | | 1.00 | | | 1.00 | | |
| Male | 0.919 | 0.800-1.055 | 0.228 | 0.876 | 0.723-1.060 | 0.174 | 0.998 | 0.829-1.203 | 0.986 | 0.962 | 0.745-1.241 | 0.763 |
| BMI | 1.019 | 1.008-1.030 | 0.001 | 1.020 | 1.004-1.035 | 0.012 | 1.035 | 1.021-1.050 | <0.001 | 1.027 | 1.007-1.047 | 0.009 |
| Approach | | | | | | | | | | | | |
| Open | 1.00 | | | 1.00 | | | 1.00 | | | 1.00 | | |
| Robotic | 0.203 | 0.161-0.256 | <0.001 | 0.206 | 0.150-0.282 | <0.001 | 0.203 | 0.146-0.283 | <0.001 | 0.173 | 0.106-0.283 | <0.001 |
| Smoking | | | | | | | | | | | | |
| No | 1.00 | | | 1.00 | | | 1.00 | | | 1.00 | | |
| Yes | 1.501 | 1.240-1.816 | <0.001 | 1.464 | 1.104-1.941 | 0.008 | 1.746 | 1.371-2.224 | <0.001 | 1.810 | 1.287-2.546 | 0.001 |
| Diabetes Mellitus | | | | | | | | | | | | |
| No | 1.00 | | | 1.00 | | | 1.00 | | | 1.00 | | |
| Yes | 1.436 | 1.226-1.681 | <0.001 | 1.073 | 0.862-1.334 | 0.529 | 1.250 | 1.008-1.550 | 0.042 | 0.965 | 0.717-1.299 | 0.816 |
| ASA | 1.916 | 1.674-2.192 | <0.001 | 1.293 | 1.055-1.586 | 0.013 | 1.591 | 1.332-1.900 | <0.001 | 1.221 | 0.932-1.599 | 0.147 |
| Albumin | | | | | | | | | | | | |
| ≥3.5 | 1.00 | | | 1.00 | | | 1.00 | | | 1.00 | | |
| <3.5 | 2.278 | 1.650-3.143 | <0.001 | 1.698 | 1.198-2.408 | 0.003 | 1.683 | 1.077-2.630 | 0.022 | 1.300 | 0.812-2.080 | 0.275 |
| CHF | | | | | | | | | | | | |
| No | 1.00 | | | 1.00 | | | 1.00 | | | 1.00 | | |
| Yes | 2.313 | 1.715-3.119 | <0.001 | 1.582 | 1.047-2.391 | 0.029 | 1.684 | 1.113-2.547 | 0.014 | 1.563 | 0.909-2.688 | 0.107 |
| COPD | | | | | | | | | | | | |
| No | 1.00 | | | 1.00 | | | 1.00 | | | 1.00 | | |
| Yes | 2.320 | 1.783-3.019 | <0.001 | 1.693 | 1.164-2.462 | 0.006 | 2.018 | 1.430-2.846 | <0.001 | 1.182 | 0.697-2.003 | 0.535 |
| Operative time | 1.004 | 1.003-1.005 | <0.001 | 1.004 | 1.003-1.005 | <0.001 | 1.003 | 1.002-1.004 | <0.001 | 1.003 | 1.002-1.004 | <0.001 |

BMI, Body mass index; ASA, American Society of Anesthesiologists; OR, Odds Ratio; 95%CI, 95% confidence interval.

## Limitations and future directions

This study has several inherent limitations due to the use of the ACS NSQIP database. First, NSQIP captures only 30-day postoperative outcomes, which precludes assessment of long-term results such as chronic pain, quality of life, and hernia recurrence, a critical endpoint when comparing surgical approaches for ventral hernia repair. Therefore, this study is unable to evaluate long-term recurrence. Lima et al. found no significant differences in hernia recurrence between robotic and open approaches at 1-year follow-up (17.1% vs 13.7%, respectively), although available data remain limited [29]. Second, the NSQIP lacks granularity in procedure-specific variables. Important factors such as hernia defect size, defect complexity, mesh use and location, and fascial closure technique are not available, limiting the ability to perform detailed subgroup analyses. Additionally, surgeon-level variables cannot be assessed, as the database does not include surgeon identifiers, precluding evaluation of the impact of surgeon experience or technical variability. Third, our analysis relies on accurate procedural coding (CPT/ICD) to identify component separation cases, which may lead to misclassification. Notably, CPT code 15734 does not differentiate between anterior and posterior techniques. While the open approach can

| Operative complications | | | | | | Medical complications | | | | | |
| Univariate | | | Multivariate | | | Univariate | | | Multivariate | | |
| OR | 95%CI | P value | OR | 95%CI | P value | OR | 95%CI | P value | OR | 95%CI | P value |
|---|---|---|---|---|---|---|---|---|---|---|---|
| 1.025 | 1.017-1.033 | <0.001 | 1.025 | 1.013-1.037 | <0.001 | 1.005 | 0.997-1.014 | 0.218 | 1.010 | 0.996-1.024 | 0.162 |
| | | | | | | | | | | | |
| 1.00 | | | 1.00 | | | 1.00 | | | 1.00 | | |
| 0.888 | 0.739-1.068 | 0.207 | 0.920 | 0.720-1.177 | 0.508 | 1.072 | 0.863-1.333 | 0.529 | 0.882 | 0.652-1.194 | 0.416 |
| 1.018 | 1.003-1.032 | 0.016 | 1.027 | 1.008-1.047 | 0.006 | 0.998 | 0.981-1.016 | 0.859 | 0.997 | 0.973-1.022 | 0.808 |
| | | | | | | | | | | | |
| 1.00 | | | 1.00 | | | 1.00 | | | 1.00 | | |
| 0.258 | 0.191-0.347 | <0.001 | 0.284 | 0.193-0.419 | <0.001 | 0.271 | 0.190-0.386 | <0.001 | 0.251 | 0.149-0.424 | <0.001 |
| | | | | | | | | | | | |
| 1.00 | | | 1.00 | | | 1.00 | | | 1.00 | | |
| 1.285 | 0.993-1.663 | 0.057 | 1.227 | 0.840-1.792 | 0.289 | 1.263 | 0.929-1.718 | 0.136 | 0.125 | 0.709-1.786 | 0.617 |
| | | | | | | | | | | | |
| 1.00 | | | 1.00 | | | 1.00 | | | 1.00 | | |
| 1.624 | 1.327-1.987 | <0.001 | 1.103 | 0.840-1.448 | 0.481 | 1.326 | 1.035-1.701 | 0.026 | 0.958 | 0.676-1.357 | 0.808 |
| 2.407 | 1.996-2.902 | <0.001 | 1.351 | 1.031-1.772 | 0.029 | 1.818 | 1.470-2.249 | <0.001 | 1.312 | 0.949-1.814 | 0.101 |
| | | | | | | | | | | | |
| 1.00 | | | 1.00 | | | 1.00 | | | 1.00 | | |
| 2.229 | 1.503-3.307 | <0.001 | 1.519 | 0.996-2.318 | 0.052 | 1.842 | 1.108-3.063 | 0.019 | 1.365 | 0.804-2.318 | 0.250 |
| | | | | | | | | | | | |
| 1.00 | | | 1.00 | | | 1.00 | | | 1.00 | | |
| 2.909 | 2.057-4.113 | <0.001 | 1.621 | 0.998-2.633 | 0.051 | 1.945 | 1.234-3.063 | 0.004 | 1.173 | 0.604-2.278 | 0.637 |
| | | | | | | | | | | | |
| 1.00 | | | 1.00 | | | 1.00 | | | 1.00 | | |
| 2.431 | 1.763-3.353 | <0.001 | 1.695 | 1.090-2.634 | 0.019 | 1.720 | 1.132-2.614 | 0.011 | 1.452 | 0.823-2.563 | 0.198 |
| 1.004 | 1.004-1.005 | <0.001 | 1.005 | 1.004-1.006 | <0.001 | 1.004 | 1.003-1.005 | <0.001 | 1.004 | 1.003-1.005 | <0.001 |

involve either technique, the robotic approach is almost exclusively posterior. This distinction is important, as the posterior technique generally results in fewer wound complications due to avoidance of large skin flaps.

Furthermore, the ACS NSQIP database does not include cost-related variables; therefore, the economic impact of robotic versus open component separation could not be evaluated in this study. However, prior studies based on smaller institutional cohorts have suggested that the robotic approach may be similarly or even more cost-effective in selected settings. Although robotic procedures are associated with higher operative costs, these may be partially offset by shorter hospital length of stay and lower rates of readmission and reintervention [30–32]. Nevertheless, dedicated cost-effectiveness analyses incorporating both clinical and economic outcomes are needed to better define the value of robotic approaches.

Finally, the NSQIP does not capture patient-reported outcomes such as postoperative pain or quality of life, limiting assessment of patient-centered endpoints. Future studies should evaluate long-term outcomes beyond the 30-day postoperative period, ideally through prospective or registry-based designs. Incorporating patient-reported outcomes and detailed operative variables would further clarify which patients benefit most from robotic versus open component separation.

**Table 4. Univariate and multivariate analysis for overall, medical, wound, and surgical complications after propensity score matching, 2022-2023.**

| Characteristics | Overall complications | | | | | | Wound complications | | | | | |
|---|---|---|---|---|---|---|---|---|---|---|---|---|
| | Univariate | | | Multivariate | | | Univariate | | | Multivariate | | |
| | OR | 95%CI | P value | OR | 95%CI | P value | OR | 95%CI | P value | OR | 95%CI | P value |
| Age - years | 1.008 | 1.002-1.015 | 0.006 | 1.007 | 0.998-1.017 | 0.147 | 0.994 | 0.986-1.002 | 0.127 | 0.991 | 0.979-1.003 | 0.157 |
| Sex | | | | | | | | | | | | |
| Female | 1.00 | | | 1.00 | | | 1.00 | | | 1.00 | | |
| Male | 0.984 | 0.845-1.146 | 0.835 | 0.898 | 0.727-1.111 | 0.322 | 1.088 | 0.886-1.336 | 0.420 | 1.019 | 0.769-1.352 | 0.894 |
| BMI | 1.019 | 1.007-1.031 | 0.002 | 1.013 | 0.996-1.031 | 0.130 | 1.036 | 1.020-1.052 | <0.001 | 1.027 | 1.005-1.050 | 0.017 |
| Approach | | | | | | | | | | | | |
| Open | 1.00 | | | 1.00 | | | 1.00 | | | 1.00 | | |
| Robotic | 0.206 | 0.163-0.261 | <0..001 | 0.193 | 0.140-0.265 | <0.001 | 0.201 | 0.144-0.282 | <0.001 | 0.164 | 0.100-0.269 | <0.001 |
| Smoking | | | | | | | | | | | | |
| No | 1.00 | | | 1.00 | | | 1.00 | | | 1.00 | | |
| Yes | 1.476 | 1.183-1.842 | 0.001 | 1.631 | 1.103-2.126 | 0.011 | 1.663 | 1.256-2.203 | <0.001 | 1.843 | 1.236-2.747 | 0.003 |
| Diabetes Mellitus | | | | | | | | | | | | |
| No | 1.00 | | | 1.00 | | | 1.00 | | | 1.00 | | |
| Yes | 1.324 | 1.098-1.596 | 0.003 | 0.963 | 0.739-1.255 | 0.780 | 1.221 | 0.948-1.573 | 0.121 | 0.813 | 0.564-1.173 | 0.269 |
| ASA | 1.928 | 1.662-2.236 | <0.001 | 1.375 | 1.100-1.719 | 0.005 | 1.708 | 1.404-2.078 | <0.001 | 1.324 | 0.983-1.784 | 0.064 |
| Albumin | | | | | | | | | | | | |
| ≥3.5 | 1.00 | | | 1.00 | | | 1.00 | | | 1.00 | | |
| <3.5 | 2.813 | 1.887-4.195 | <0.001 | 2.332 | 1.500-3.625 | <0.001 | 2.436 | 1.460-4.064 | 0.001 | 2.052 | 1.185-3.552 | 0.010 |
| CHF | | | | | | | | | | | | |
| No | 1.00 | | | 1.00 | | | 1.00 | | | 1.00 | | |
| Yes | 2.587 | 1.840-3.635 | <0.001 | 1.959 | 1.224-3.134 | 0.005 | 2.027 | 1.289-3.187 | 0.002 | 0.944 | 0.490-1.818 | 0.863 |
| COPD | | | | | | | | | | | | |
| No | 1.00 | | | 1.00 | | | 1.00 | | | 1.00 | | |
| Yes | 2.050 | 1.507-2.790 | <0.001 | 1.598 | 1.023-2.495 | 0.039 | 1.848 | 1.234-2.767 | 0.003 | 0.944 | 0.490-1.818 | 0.863 |
| Operative time | 1.004 | 1.003-1.005 | <0.001 | 1.004 | 1.003-1.005 | <0.001 | 1.003 | 1.002-1.004 | <0.001 | 1.003 | 1.002-1.004 | <0.001 |

BMI, Body mass index; ASA, American Society of Anesthesiologists; COPD, chronic obstructive pulmonary disease; CHF, congestive heart failure; OR, Odds Ratio; 95%CI, 95% confidence interval.

## Conclusions

Robotic component separation for ventral hernia repair was associated with lower postoperative complication rates, shorter hospital stays, and fewer readmissions compared with the open approach. These benefits remained significant after multivariable analysis and propensity score matching, supporting the use of the robotic technique when possible. However, both techniques have value, and the choice should be made on a case-by-case basis, taking into account patient-specific factors and the surgeon's experience and expertise with each approach.

While our findings highlight the robotic approach as a promising and effective strategy, important limitations remain, including the absence of long-term outcomes such as hernia recurrence rates and patient-reported measures. Although randomized controlled trials would validate the overall evidence, additional studies, including evaluations of long-term durability and cost-effectiveness, are needed to better inform surgical decision-making.

| Operative complications | | | | | | Medical complications | | | | | |
| Univariate | | | Multivariate | | | Univariate | | | Multivariate | | |
| OR | 95%CI | P value | OR | 95%CI | P value | OR | 95%CI | P value | OR | 95%CI | P value |
|---|---|---|---|---|---|---|---|---|---|---|---|
| 1.022 | 1.014-1.031 | <0.001 | 1.022 | 1.009-1.035 | 0.001 | 1.004 | 0.994-1.013 | 0.443 | 1.006 | 0.991-1.020 | 0.462 |
| | | | | | | | | | | | |
| 1.00 | | | 1.00 | | | 1.00 | | | 1.00 | | |
| 0.940 | 0.766-1.152 | 0.549 | 0.927 | 0.706-1.219 | 0.589 | 1.087 | 0.855-1.382 | 0.494 | 0.883 | 0.635-1.228 | 0.460 |
| 1.014 | 0.998-1.030 | 0.092 | 1.013 | 0.991-1.036 | 0.260 | 1.006 | 0.987-1.025 | 0.532 | 1.000 | 0.974-1.028 | 0.989 |
| | | | | | | | | | | | |
| 1.00 | | | 1.00 | | | 1.00 | | | 1.00 | | |
| 0.265 | 0.196-0.358 | <0.001 | 0.271 | 0.182-0.402 | <0.001 | 0.269 | 0.188-0.385 | <0.001 | 0.229 | 0.135-0.389 | <0.001 |
| | | | | | | | | | | | |
| 1.00 | | | 1.00 | | | 1.00 | | | 1.00 | | |
| 1.441 | 1.077-1.927 | 0.014 | 1.598 | 1.049-2.435 | 0.029 | 1.152 | 0.800-1660 | 0.446 | 0.972 | 0.552-1.712 | 0.923 |
| | | | | | | | | | | | |
| 1.00 | | | 1.00 | | | 1.00 | | | 1.00 | | |
| 1.497 | 1.177-1.904 | 0.001 | 1.069 | 0.770-1.484 | 0.688 | 1.350 | 1.012-1.800 | 0.041 | 0.911 | 0.600-1.383 | 0.661 |
| 2.339 | 1.906-2.871 | <0.001 | 1.435 | 1.065-1.934 | 0.018 | 1.743 | 1.385-2.194 | <0.001 | 1.394 | 0.981-1.980 | 0.064 |
| | | | | | | | | | | | |
| 1.00 | | | 1.00 | | | 1.00 | | | 1.00 | | |
| 2.813 | 1.746-4.530 | <0.001 | 2.086 | 1.244-3.500 | 0.005 | 2.317 | 1.270-4.226 | 0.006 | 1.905 | 1.015-3.574 | 0.045 |
| | | | | | | | | | | | |
| 1.00 | | | 1.00 | | | 1.00 | | | 1.00 | | |
| 3.171 | 2.135-4.708 | <0.001 | 1.827 | 1.048-3.196 | 0.035 | 2.209 | 1.334-3.659 | 0.002 | 1.165 | 0.537-2.528 | 0.699 |
| | | | | | | | | | | | |
| 1.00 | | | 1.00 | | | 1.00 | | | 1.00 | | |
| 2.058 | 1.396-3.034 | <0.001 | 1.424 | 0.828-2.447 | 0.201 | 1.345 | 0.798-2.269 | 0.266 | 1.055 | 0.505-2.203 | 0.886 |
| 1.005 | 1.004-1.005 | <0.001 | 1.005 | 1.004-1.006 | <0..001 | 1.004 | 1.003-1.005 | <0.001 | 1.003 | 1.002-1.005 | <0.001 |

## Supporting information

**S1 File.**

(XLSX)

## Acknowledgments

The authors thank the Universidad Científica del Sur for their support in the publication of this research/project.

## Author contributions

**Conceptualization:** Silvana Maldonado, Carlos Gonzales, Anshumi Desai, Jiddu Guart, Alba Zevallos Ventura, Bryan Valcarcel, Joseph Escandon, Natalia Mejia, Camila Franco Mesa, J. Smith Torres Roman, Jose Luis Barrueto, Hamed Sarikhani, Gabriel De la Cruz Ku, Donald Czerniach, Sameer Patel, Adam Walchak.

**Data curation:** Silvana Maldonado, Carlos Gonzales, Anshumi Desai, Jiddu Guart, Alba Zevallos Ventura, Bryan Valcarcel, Joseph Escandon, Natalia Mejia, Camila Franco Mesa, J. Smith Torres Roman, Jose Luis Barrueto, Hamed Sarikhani, Gabriel De la Cruz Ku, Donald Czerniach, Sameer Patel, Adam Walchak.

**Formal analysis:** Carlos Gonzales, Anshumi Desai, Jiddu Guart, Alba Zevallos Ventura, Bryan Valcarcel, Camila Franco Mesa, Jose Luis Barrueto, Hamed Sarikhani, Gabriel De la Cruz Ku, Donald Czerniach, Sameer Patel, Adam Walchak.

**Funding acquisition:** Carlos Gonzales, Bryan Valcarcel, Donald Czerniach, Sameer Patel, Adam Walchak.

**Investigation:** Silvana Maldonado, Carlos Gonzales, Anshumi Desai, Jiddu Guart, Alba Zevallos Ventura, Bryan Valcarcel, Joseph Escandon, Natalia Mejia, Camila Franco Mesa, J. Smith Torres Roman, Jose Luis Barrueto, Hamed Sarikhani, Gabriel De la Cruz Ku, Donald Czerniach, Sameer Patel, Adam Walchak.

**Methodology:** Silvana Maldonado, Carlos Gonzales, Anshumi Desai, Jiddu Guart, Alba Zevallos Ventura, Joseph Escandon, Natalia Mejia, Camila Franco Mesa, J. Smith Torres Roman, Jose Luis Barrueto, Hamed Sarikhani, Gabriel De la Cruz Ku, Donald Czerniach, Sameer Patel, Adam Walchak.

**Project administration:** Silvana Maldonado, Camila Franco Mesa, J. Smith Torres Roman, Gabriel De la Cruz Ku, Donald Czerniach, Sameer Patel, Adam Walchak.

**Resources:** Carlos Gonzales, Anshumi Desai, Joseph Escandon, Natalia Mejia, Jose Luis Barrueto, Gabriel De la Cruz Ku, Sameer Patel, Adam Walchak.

**Software:** Silvana Maldonado, Carlos Gonzales, Anshumi Desai, Jiddu Guart, Alba Zevallos Ventura, Joseph Escandon, Camila Franco Mesa, Hamed Sarikhani, Gabriel De la Cruz Ku, Adam Walchak.

**Supervision:** Silvana Maldonado, Anshumi Desai, Jiddu Guart, Alba Zevallos Ventura, Joseph Escandon, Natalia Mejia, Camila Franco Mesa, J. Smith Torres Roman, Jose Luis Barrueto, Hamed Sarikhani, Gabriel De la Cruz Ku, Donald Czerniach, Sameer Patel, Adam Walchak.

**Validation:** Silvana Maldonado, Carlos Gonzales, Alba Zevallos Ventura, Bryan Valcarcel, J. Smith Torres Roman, Jose Luis Barrueto, Gabriel De la Cruz Ku, Donald Czerniach, Sameer Patel, Adam Walchak.

**Visualization:** Silvana Maldonado, Carlos Gonzales, Jiddu Guart, Bryan Valcarcel, Joseph Escandon, Natalia Mejia, Camila Franco Mesa, J. Smith Torres Roman, Jose Luis Barrueto, Hamed Sarikhani, Gabriel De la Cruz Ku, Donald Czerniach, Sameer Patel, Adam Walchak.

**Writing – original draft:** Silvana Maldonado, Carlos Gonzales, Anshumi Desai, Jiddu Guart, Alba Zevallos Ventura, Bryan Valcarcel, Joseph Escandon, Natalia Mejia, Camila Franco Mesa, J. Smith Torres Roman, Jose Luis Barrueto, Hamed Sarikhani, Gabriel De la Cruz Ku, Donald Czerniach, Sameer Patel, Adam Walchak.

**Writing – review & editing:** Silvana Maldonado, Carlos Gonzales, Anshumi Desai, Jiddu Guart, Alba Zevallos Ventura, Bryan Valcarcel, Joseph Escandon, Natalia Mejia, Camila Franco Mesa, J. Smith Torres Roman, Jose Luis Barrueto, Hamed Sarikhani, Gabriel De la Cruz Ku, Donald Czerniach, Sameer Patel, Adam Walchak.

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
