## [Decision Letter · Decision Letter 0]

26 Jan 2026

Dear Dr. de la Cruz-Ku,

Thank you for submitting your manuscript to PLOS ONE. After careful consideration, we feel that it has merit but does not fully meet PLOS ONE’s publication criteria as it currently stands. Therefore, we invite you to submit a revised version of the manuscript that addresses the points raised during the review process.

We look forward to receiving your revised manuscript.

Kind regards,

Shekhar Gogna

Academic Editor

PLOS One

Journal Requirements:

“The study was funded by the authors, and no financial support was received from any financial institution or external entity.”

Additional Editor Comments:

I personally liked the paper a lot, well written and I would like to congratulate you all. The Reviewer 1 has raised some minor issue and i would appreciate if you all can address that. I don't necessarily agree with reviewer 2. Reviewer 3 has accidently put in the review for some random unrelated manuscript we will try and fix that. But I don't want to delay this further and once the minor revisions are done, we can move forward.

Again, great job !

Reviewers' comments:

Reviewer's Responses to Questions

**Comments to the Author**

1. Is the manuscript technically sound, and do the data support the conclusions?

Reviewer #1: Yes

Reviewer #2: Yes

Reviewer #3: Yes

2. Has the statistical analysis been performed appropriately and rigorously?

Reviewer #1: Yes

Reviewer #2: Yes

Reviewer #3: Yes

3. Have the authors made all data underlying the findings in their manuscript fully available?

Reviewer #1: Yes

Reviewer #2: Yes

Reviewer #3: No

4. Is the manuscript presented in an intelligible fashion and written in standard English?

Reviewer #1: Yes

Reviewer #2: Yes

Reviewer #3: Yes

Reviewer #1: I would like to congratulate the authors on conducting this important nationwide analysis of complex ventral hernia using the NSQIP. This dataset is one of the largest surgical databases and represents patients within the US. Even though the authors have done a commendable job in assembling and analyzing a large cohort, I have a few questions, methodological clarifications, and suggestions. Addressing these would help strengthen the manuscript’s clarity, reproducibility, and interpretability, thereby improving its value to the readership.

Methods:

1. Does CPT code 15734 specify posterior or anterior component separation? This distinction is important because anterior component separation is associated with a higher complication rate than posterior component separation, and a robotic approach inherently involves only posterior component separation.

2. Does the ICD-10 code of K43.9 include patient with incisional or spontaneous ventral hernias? Did the author consider using ICD-10 code of K43.2 (Incisional hernia)?

3. In the Definition of variables and classification of cohorts” paragraph, author stated, “Patients were categorized into two cohorts based on the surgical approach: standard open ventral hernia repair versus ventral hernia repair with component separation (open or laparoscopic)” Does the author mean open vs robotic or laparoscopic?

4. Is there a variable in NSQIP to determine whether the procedure was done robotic or laparoscopic. CPT code of 49659 is classified as "unlisted laparoscopy procedure.

5. Under statistical analysis section, the author stated Patients were matched on age, sex, smoking status, body mass index, diabetes mellitus, and serum albumin level. Based on Table. 1, BMI was statistically different between the two groups. Please correct the statement.

Results:

1. Under results section, the author stated “after propensity score matching, which achieved balance across age, albumin, BMI, diabetes mellitus, sex, and smoking status, robotic surgery remained strongly associated with reduced complications (Figure 1)”. The BMI was different in both groups based on Table 1.

Conclusions:

1. The authors concluded that “robotic component separation for complex ventral hernia repair was associated with lower postoperative complication rates, shorter hospital stays, and fewer readmissions compared with the open approach. I would recommend removing the term complex, as NSQIP does not include a hernia size variable, unless the indication for component separation itself is being used to define complexity.

Reviewer #2: The authors present an interesting topic analyzing the ACS-NSQIP database to look at robotic vs open component separation techniques. They found that the robotic technique has significantly improved postoperative outcomes vs open.

The main limitation of this study is the same with any large database set which is that this data is not granular enough to draw clinically relevant conclusions. The main outcome that needs to be studied with this study is recurrence rate of hernia between the two techniques. If the immediate perioperative outcomes of the open technique are inferior but the hernia recurrence rates and hernia related complication rates are lower, this would help surgeons make a much more informed decision regarding surgical technique choice.

Subgroup analysis of the different meshes used vs no mesh, and the location of mesh would greatly improve the power of this study.

What about the economic costs of buying/utilizing robotic equipment vs. open?

Reviewer #3: I read with interest your study entitled “Clinical Outcomes in Preemptive versus Non-preemptive Simultaneous Pancreas–Kidney Transplant Recipients: A Systematic Review and Meta-analysis.”

This systematic review and meta-analysis includes seven retrospective comparative studies evaluating preemptive simultaneous pancreas–kidney transplantation (pSPKT) versus non-preemptive simultaneous pancreas–kidney transplantation (nSPKT). The authors report that pSPKT is associated with superior long-term patient survival and kidney graft survival compared with nSPKT.

Overall, the study is well structured and clearly written. The methodology appears sound, and the focused results are reported appropriately. The novelty of the study is acceptable; however, the clinical message warrants further scrutiny.

The main concern is that the compared groups may not be truly comparable. It is possible that patients undergoing pSPKT received kidneys from living donors, as achieving SPKT in a preemptive setting without a living donor is often challenging. This factor alone could largely explain the observed differences in outcomes and therefore should be discussed in detail.

Furthermore, although some results reach statistical significance, the clinical relevance of these differences appears limited, as the reported risk ratios are very similar between groups. This issue should also be discussed more thoroughly, particularly in the context of the ongoing organ shortage in many countries, where patients may remain on waiting lists for pancreas–kidney transplantation for years without ever receiving the needed organs.

Minor comments:

Abstract:

Results should be reported in a neutral manner. The use of terms such as “slightly better” when results favor nSPKT, versus “significantly higher” or “favored pSPKT” when results favor pSPKT, is inconsistent—especially since the risk ratios and 95% confidence intervals are in a similar range.

Discussion:

The following sentence is unclear and requires clarification regarding the comparison being made:

“For example, pretransplant dialysis of more than 1 year was associated with a substantially higher mortality risk in SPKT patients with T2DM than in those with T1DM (a 160% vs. 59% higher risk, respectively) compared to those who underwent SPKT in a preemptive setting.”

It is not clear whether this comparison refers to T1DM versus T2DM, or preemptive versus non-preemptive SPKT, and this should be explicitly clarified.

.

Reviewer #1: No

Reviewer #2: No

Reviewer #3: **Yes:** Sepehr Abbasi DezfouliSepehr Abbasi DezfouliSepehr Abbasi DezfouliSepehr Abbasi Dezfouli

---

## [Author Response · Author response to Decision Letter 1]

22 Mar 2026

RESPONSE TO REVIEWERS

Reviewer 1

Reviewer’s comment:

I would like to congratulate the authors on conducting this important nationwide analysis of complex ventral hernia using the NSQIP. This dataset is one of the largest surgical databases and represents patients within the US. Even though the authors have done a commendable job in assembling and analyzing a large cohort, I have a few questions, methodological clarifications, and suggestions. Addressing these would help strengthen the manuscript’s clarity, reproducibility, and interpretability, thereby improving its value to the readership.

Authors’ response:

We thank the reviewer for this insightful comment.

Reviewer’s comment:

Methods: 1. Does CPT code 15734 specify posterior or anterior component separation? This distinction is important because anterior component separation is associated with a higher complication rate than posterior component separation, and a robotic approach inherently involves only posterior component separation.

Authors’ response:

We thank the reviewer for this important comment. We acknowledge that CPT code 15734 (muscle–fascial flap) encompasses both anterior and posterior component separation techniques. However, the NSQIP database does not include specific modifiers that allow differentiation between these approaches. While the open cohort may include both anterior and posterior component separation, the robotic approach predominantly involves posterior component separation, most commonly transversus abdominis release (TAR). To improve clarity, we have revised the Methods section to specify the scope of this CPT code and have also included this limitation in the Discussion section.

Reviewer’s comment:

Methods: 2. Does the ICD-10 code of K43.9 include patient with incisional or spontaneous ventral hernias? Did the author consider using ICD-10 code of K43.2 (Incisional hernia)?

Authors’ response:

We thank the reviewer for this important comment. To ensure a comprehensive identification of complex cases, a double-coding strategy was employed. ICD-10 codes K43.2 (incisional hernia without obstruction or gangrene) and K43.9 (unspecified ventral hernia) were included in our query to capture a broad population of patients with ventral wall defects, including both incisional and spontaneous ventral hernias. Following the reviewer’s comment, we re-evaluated the dataset and carefully verified the coding strategy, confirming that both ICD-10 codes (K43.2 and K43.9) were appropriately included in the analysis. This has now been clarified and corrected in the Materials and Methods section of the manuscript.

Cases involving component separation were then identified using CPT codes 15734 and 49659; for the latter, the NSQIP robotic-assisted variable was used to confirm the robotic approach. This strategy allowed the study population to be defined not only by the hernia diagnosis but also by the corresponding surgical procedure. It should be noted that the available coding does not differentiate between anterior and posterior component separation techniques.

Reviewer’s comment:

Methods: 3. In the Definition of variables and classification of cohorts” paragraph, author stated, “Patients were categorized into two cohorts based on the surgical approach: standard open ventral hernia repair versus ventral hernia repair with component separation (open or laparoscopic)” Does the author mean open vs robotic or laparoscopic?

Authors’ response:

We thank the reviewer for this observation. We agree that the description of the primary exposure in the “Definition of variables and classification of cohorts” section was unclear, and we apologize for this oversight. To clarify, our analysis compares two cohorts based on the surgical approach used for component separation: open versus robotic. The laparoscopic approach was not included in this study. We have revised the wording in the Methods section to accurately reflect this cohort definition and eliminate any ambiguity. We appreciate the reviewer’s comment, which helped us improve the clarity of the manuscript.

Reviewer’s comment:

Methods 4. Is there a variable in NSQIP to determine whether the procedure was done robotic or laparoscopic. CPT code of 49659 is classified as "unlisted laparoscopy procedure.

Authors’ response:

We thank the reviewer for this important comment. We acknowledge that CPT code 49659 is classified as an “unlisted laparoscopy procedure.” However, in the NSQIP Participant Use Data File (PUF) for 2022 and 2023, there is a specific variable indicating whether a procedure was performed using robotic assistance. For this reason, our study cohort was limited to these years, allowing us to accurately identify robotic-assisted cases and exclude conventional laparoscopic procedures. We have revised the Methods section to clearly describe how robotic cases were identified using this variable. We appreciate the reviewer’s comment, which helped us clarify this aspect of our methodology.

Reviewer’s comment:

Methods 5. Under statistical analysis section, the author stated Patients were matched on age, sex, smoking status, body mass index, diabetes mellitus, and serum albumin level. Based on Table. 1, BMI was statistically different between the two groups. Please correct the statement.

Authors’ response:

We thank the reviewer for this important observation. After re-evaluating the matched cohorts, we identified that the BMI value in Table 1 required correction. Following propensity score matching, BMI was no longer significantly different between groups (p = 0.289), indicating appropriate balance between the cohorts. We have corrected this value in Table 1 and updated the corresponding statement in the Methods and Results sections to accurately reflect the post-matching balance between groups. We appreciate the reviewer’s comment, which helped us identify and correct this issue.

Reviewer’s comment:

Results: 1. Under results section, the author stated “after propensity score matching, which achieved balance across age, albumin, BMI, diabetes mellitus, sex, and smoking status, robotic surgery remained strongly associated with reduced complications (Figure 1)”. The BMI was different in both groups based on Table 1.

Authors’ response:

We thank the reviewer for this observation. After re-evaluating the matched cohorts, we found that the BMI value in Table 1 required correction. Following this correction, BMI was not significantly different between the groups after propensity score matching (p = 0.289), confirming adequate balance across the matched variables. Table 1 has been updated, and the Results section has been revised accordingly to accurately reflect the post-matching balance between groups.

Reviewer’s comment:

Conclusions: 1. The authors concluded that “robotic component separation for complex ventral hernia repair was associated with lower postoperative complication rates, shorter hospital stays, and fewer readmissions compared with the open approach. I would recommend removing the term complex, as NSQIP does not include a hernia size variable, unless the indication for component separation itself is being used to define complexity.

Authors’ response:

We thank the reviewer for this valuable comment. We acknowledge that the ACS NSQIP database does not include hernia size as a variable. While the use of component separation techniques is often associated with more complex ventral hernia repairs, we agree that the use of the term “complex” is not methodologically supported by the available database variables in this study. Therefore, we have removed this term from the conclusion and revised the manuscript accordingly. We appreciate the reviewer’s suggestion, which helped improve the accuracy of our wording.

Reviewer 2

Reviewer’s comment:

The authors present an interesting topic analyzing the ACS-NSQIP database to look at robotic vs open component separation techniques. They found that the robotic technique has significantly improved postoperative outcomes vs open.

Authors’ response:

We thank the reviewer for the positive feedback and for acknowledging the relevance of our study evaluating robotic versus open component separation techniques using the ACS NSQIP database.

Reviewer’s comment:

The main limitation of this study is the same with any large database set which is that this data is not granular enough to draw clinically relevant conclusions. The main outcome that needs to be studied with this study is recurrence rate of hernia between the two techniques. If the immediate perioperative outcomes of the open technique are inferior but the hernia recurrence rates and hernia related complication rates are lower, this would help surgeons make a much more informed decision regarding surgical technique choice.

Subgroup analysis of the different meshes used vs no mesh, and the location of mesh would greatly improve the power of this study.

Authors’ response:

We thank the reviewer for this insightful comment. We agree that one of the main limitations of large administrative or clinical databases such as ACS NSQIP is the lack of granular clinical information. Important variables such as hernia size, mesh use, mesh type, and mesh location are not captured in this dataset, which prevents performing subgroup analyses based on these factors. Additionally, the ACS NSQIP database only records outcomes within 30 days postoperatively, and therefore long-term outcomes such as hernia recurrence or hernia-related complications cannot be assessed. We agree that these outcomes are critical when comparing surgical techniques.

To address this limitation, we have now explicitly added this point to the limitations section of the manuscript and incorporated available evidence from the literature. Notably, prior studies such as Lima et al. have reported no significant differences in hernia recurrence between robotic and open approaches at 1-year follow-up (17.1% vs 13.7%, respectively), although long-term data remain limited.

We agree that further studies with longer follow-up and more granular clinical data, including mesh characteristics and hernia complexity, are needed to better inform surgical decision-making.

Reviewer’s comment:

What about the economic costs of buying/utilizing robotic equipment vs. open?

Authors’ response:

We thank the reviewer for this important comment. We agree that the economic implications of robotic surgery are an important consideration when comparing surgical approaches. However, the ACS NSQIP database does not include cost-related variables, which precludes performing a formal cost analysis in the present study.

Nonetheless, prior studies based on smaller institutional cohorts have evaluated the economic impact of robotic ventral hernia repair and have suggested that the robotic approach may be similarly or even more cost-effective in certain settings. Although robotic procedures are associated with higher operative costs in the operating room, these costs may be partially offset by shorter hospital length of stay and lower rates of readmission and reintervention within 90 days. We have added a statement acknowledging this point in the Discussion section. Further studies specifically designed to evaluate the economic impact of robotic versus open approaches are warranted.

Editor Comments

Editor’s comment:

I personally liked the paper a lot, well written and I would like to congratulate you all. The Reviewer 1 has raised some minor issue and i would appreciate if you all can address that. I don't necessarily agree with reviewer 2. Reviewer 3 has accidently put in the review for some random unrelated manuscript we will try and fix that. But I don't want to delay this further and once the minor revisions are done, we can move forward.

Again, great job !

Authors’ response:

Thank you very much for your kind words. We truly appreciate your feedback. We have carefully addressed the minor issues raised by Reviewer 1 and have revised the manuscript accordingly. We understand your comments regarding Reviewer 2 and 3, and we are grateful for the clarifications provided. We look forward to the next steps and hope that the revised manuscript meets the requirements for publication.

---

## [Editor Report · Decision Letter 1]

8 Apr 2026

Robotic versus Open Component Separation: A Retrospective Cohort and Propensity Score Analysis of Complication Rates and Clinical Outcomes

PONE-D-25-66542R1

Dear Dr. Cruz Ku,

We’re pleased to inform you that your manuscript has been judged scientifically suitable for publication and will be formally accepted for publication once it meets all outstanding technical requirements.

Kind regards,

Shekhar Gogna

Academic Editor

PLOS One

---

## [Editor Report · Acceptance letter]

PONE-D-25-66542R1

PLOS One

Dear Dr. De la Cruz Ku,

I'm pleased to inform you that your manuscript has been deemed suitable for publication in PLOS One. Congratulations! Your manuscript is now being handed over to our production team.

Kind regards,

on behalf of

Dr. Shekhar Gogna

Academic Editor

PLOS One